# *CARINH*, an Interferon-Induced LncRNA in Cancer and Inflammation

**DOI:** 10.3390/ncrna11060079

**Published:** 2025-11-21

**Authors:** Morgane Gourvest, Coen van Solingen

**Affiliations:** Department of Medicine, Cardiovascular Research Center, New York University Grossman School of Medicine, New York, NY 10016, USA; morgane.gourvest@nyulangone.org

**Keywords:** *CARINH*, long noncoding RNA, innate immunity, IRF1, interferon, cancer

## Abstract

*CARINH* is an intriguing long noncoding RNA whose unique regulatory functions intersect the seemingly distinct processes of innate immunity and cancer development. Notably, *CARINH* is conserved across species, offering powerful experimental models for uncovering its mechanistic roles and physiological functions across diverse biological contexts. Stimulated by interferons and viral infections, *CARINH* stands out as a key player in the body’s antiviral defense mechanisms. Additionally, its dysregulation has been implicated in autoimmune disorders such as psoriasis, asthma, and inflammatory bowel disease, underscoring its broader role in maintaining immune homeostasis. Furthermore, alterations in *CARINH* expression have been connected to cancer progression, highlighting its dual role in immune response and tumor suppression. In this review, we delve into *CARINH*’s pivotal function in modulating interferon responses and influencing cancer development, with a focus on the molecular pathways that regulate its expression and contribute to its diverse roles. Understanding these pathways is crucial for evaluating *CARINH*’s significance as a biomarker and therapeutic target, potentially leading to groundbreaking advancements in medical research and treatment strategies.

## 1. Introduction

In recent years, the identification and characterization of noncoding RNAs (ncRNAs) have drastically transformed our perception of genetic function, revealing the intricate ways in which genomic elements can impact cellular processes. This newfound understanding of the genome’s so-called “junk DNA” has unveiled a variety of RNA classes that are essential for modulating protein-coding genes and organizing the genome’s structural framework [1]. Within the broad spectrum of ncRNAs discovered over the last few decades, long noncoding RNAs (lncRNAs) have been particularly instrumental in illuminating RNA’s significant role in gene regulatory mechanisms [2]. Like conventional messenger RNAs (mRNAs), ncRNAs are transcribed from DNA, yet they do not encode proteins. Their synthesis is primarily mediated by RNA polymerase II or, less commonly, by RNA polymerase I and III for certain ncRNA subclasses. The initiation of ncRNA transcription is tightly regulated by promoters, enhancers, and epigenetic modifications, while transcription factors and chromatin remodeling complexes modulate ncRNA loci chromatin accessibility. Following transcription, primary ncRNA transcripts undergo similar processing steps to those of mRNAs, including capping, splicing, and 3′ polyadenylation, reflecting their shared biogenesis pathways, although many ncRNAs also exhibit alternative splicing patterns, variable polyadenylation, or incomplete processing. The stability and turnover of ncRNAs are further controlled by RNA-binding proteins, RNA modification pathways (such as m^6^A), and RNA degradation mechanisms, ensuring appropriate expression levels within the cell. Moreover, ncRNA expression is highly dynamic and responsive to physiological and pathological cues. External stimuli such as stress, infection, and metabolic alterations, as well as diverse intracellular signaling pathways, can rapidly modulate ncRNA transcription and processing, allowing cells to rapidly adapt transcriptional responses to environmental cues. Collectively, these multilayered regulatory processes integrate lncRNA biogenesis, processing, and context-dependent expression within broader transcriptional and post-transcriptional control networks [3,4].

LncRNAs, a diverse group of transcripts, are distinguished by their length of over 200 nucleotides. They exhibit limited evolutionary conservation across species, which has posed challenges to deciphering their functional complexities [5,6,7]. These molecules are versatile, residing either in the nucleus or the cytoplasm, and they fulfill distinct tasks within these cellular compartments [8,9]. In the nucleus, lncRNAs are known for their multifaceted functions and can assemble into ribonucleoprotein complexes that serve as guides or decoys, influence nuclear organization, and contribute to the formation of higher-order chromosomal structures [10,11]. Additionally, they play roles in scaffolding effector proteins, enabling the modulation—either reduction or enhancement—of gene regulatory activities [12]. Furthermore, lncRNAs operate in both *cis* and *trans* regulatory modes; they regulate gene expressions either near their transcription sites or at distal genomic locations [13,14]. Within the cytoplasm, lncRNAs act as competitive inhibitors to microRNAs (miRNAs), where miRNAs typically bind to the 3′ untranslated regions of target mRNAs to suppress their translation. By sponging miRNAs, lncRNAs can mitigate this inhibitory effect. Besides their interaction with miRNAs, cytoplasmic lncRNAs are involved in post-transcriptional regulation by affecting the stability and translation of mRNA directly [15,16,17].

Despite significant advancements in lncRNA research, our understanding of their impacts on health and disease remains incomplete. One of the principal challenges stems from the relatively poor conservation of human lncRNAs across species. This is thought to arise because regulatory sequences, including lncRNAs, evolve at a faster pace than protein-coding sequences, likely due to the more relaxed structure–function constraints these molecules face [2]. This evolutionary flexibility positions lncRNAs as intriguing components of genomic innovation. Interestingly, while many lncRNAs exhibit rapid sequence divergence, several notable examples, such as *MALAT1*, demonstrate strong conservation across vertebrates [18]. This implies that while most lncRNAs may tolerate substantial sequence variation, a subset of them retains critical functional or structural features under stronger evolutionary pressure. Research into conserved lncRNA families is still at an early stage, and further studies are needed to define how conservation relates to biological activity and physiological relevance [2,19]. The limited conversation of many human lncRNAs has restricted the predictive power of preclinical animal models, resulting in an underestimation of their contributions to disease progression and a slowdown in efforts to harness their therapeutic potential. In this context, this review is focused on the lncRNA Colitis Associated IRF1 antisense Regulator of Intestinal Homeostasis (*CARINH*), a lncRNA conserved in mice that plays important regulatory roles in interferon-driven innate immunity [20,21,22,23,24] and various cancers [25,26,27,28,29]. Remarkably, the presence of a mouse ortholog to *CARINH* has offered unique opportunities to investigate its functions in preclinical models using *Carinh* knockout mice [22,23]. By investigating *CARINH* in these in vivo models as well complementary in vitro models, researchers have gained critical insights into its biological functions and potential therapeutic applications, which we have highlighted in this review.

## 2. CARINH: Nomenclature, Genomic Location, and Synteny

The GENCODE consortium and other groups have made substantial efforts to establish a unified system for naming lncRNAs, drawing on criteria such as their genomic location relative to protein-coding genes, their broader genomic context, or predicted structural and functional features [30]. Regardless of these efforts, uniformity has yet to be fully achieved. As a result, many lncRNAs continue to be annotated under several different names across distinct databases, creating redundancy and confusion. LncRNA *CARINH* further illustrates this, as it is associated with multiple, sometimes inconsistent aliases. Currently, The Human Genome Organisation (HUGO) Gene Nomenclature Committee (HGNC) [31] lists *CARINH* as the approved short form for the full gene name ‘Colitis Associated Interferon Response Factor 1 [IRF1] antisense Regulator of Intestinal Homeostasis’, with previous gene names being ‘IRF1 Antisense RNA 1′ (*IRF1-AS*, *IRF1-AS1*) and ‘Chromosome 5 Open Reading Frame 56′ (*C5ORF56*). *CARINH* was originally identified in human hepatic Huh7 cells exposed to varying concentrations of interferon (IFN)α for up to 72 h. Within this experimental setting, researchers detected a collection of lncRNAs that were either induced (IFN-Stimulated lncRNAs) or repressed (IFN-Downregulated lncRNAs) by IFN signaling. Notably, three of these transcripts displayed expression profiles closely paralleling those of their neighboring protein-coding genes. Among these pairs was the coding gene *IRF1* and the adjacent noncoding transcript *AC116366.6* or *ENST00000461203*, which was subsequently named IFN-Stimulated lncRNA 8 (*ISR8*) [21]. Of note, *ISR8* represents one of the splice variants for *CARINH* currently annotated in the Ensembl database.

The *CARINH* gene, also designated by identifiers including *ENSG00000197536* (Ensembl v115), *NR_161242* (GenBank), *AC116366* (GENCODE v48), and *441108* (NCBI), is located on human chromosome 5. It comprises three splice variants (*ENST00000612967.2 [CARINH-V1]*, *ENST00000337752.6 [CARINH-V2]*, *and ENST00000407797.6 [CARINH-V3]*) currently documented in GENCODE, all of which have the same first three exons but vary in their 3′ end sequences [22]. *CARINH* is positioned antisense to the *IRF1* gene; one of these splice variants, *ENST00000612967.2*, partially overlaps with the coding sequence of *IRF1* [22]. Besides *IRF1*, the genetic locus in which *CARINH* is situated contains several genes that are imporant in inflammatory processes such as interleukin(*IL*)*-4*, *IL-13* and *IL-5* [32,33], as well as genes involved in the processing of DNA double-stranded breaks (*RAD50*) [34] and cellular uptake of carnitine (*SLC22A5*)^31^ (Figure 1a).

Synteny analysis comparing human and mouse genomes has identified that *CARINH* is one of the rare human lncRNAs that has a mouse ortholog, known as *Carinh*—also referred to as *Gm12216*—located on mouse chromosome 11. This mouse ortholog is similarly positioned antisense to the mouse *Irf1* gene and in genomic vicinity to *Il-4*, *Il-13*, *Il-15*, *Rad50*, and *Slc22a5* (Figure 1b) [22,23]. The presence of a mouse ortholog to *CARINH* presents a unique opportunity to explore its functions in vivo using *Carinh* knock-out mice, as demonstrated in inflammatory bowel disease and viral infection [22,23]. A deeper exploration of the regulatory functions of this specific lncRNA and its mouse ortholog, situated in a locus encompassing crucial genes for inflammatory signaling, may reveal essential mechanisms underpinning cellular responses and potentially influence clinical outcomes.

## 3. CARINH and Innate Immunity

Interferons (IFNs) are critical components of the innate immune system, primarily responsible for the initial defense against infections. These cytokines are produced and released by host cells upon sensing pathogens such as viruses, bacteria, and parasites. By binding to their specific receptors at the surface of cells, IFNs activate signaling pathways that induce the expression of IFN-stimulated genes (ISGs). These ISGs encode antiviral, antimicrobial, and immunoregulatory proteins that collectively restrict pathogen replication, promote antigen presentation, and shape downstream immune responses, enhancing the detection and elimination of pathogens. Additionally, IFNs play a role in activating and recruiting effector immune cells, including natural killer cells and macrophages, thereby orchestrating a coordinated immune response [35,36,37].

### 3.1. CARINH in Viral Infection

Initial evidence implicating *CARINH* in IFN signaling and antiviral immunity emerged from a study profiling lncRNA expression in IFNα-stimulated hepatoma cells. This work identified a signature of interferon-stimulated lncRNAs (ISRs), among which *ISR8* (a splice variant of *CARINH*) was particularly robustly induced. *CARINH* expression was found to be co-regulated with the nearby interferon-stimulated gene *IRF1*, with its induction consistently observed across various cell types in response to both IFNα and IFNβ. Additionally, *CARINH* levels were elevated in liver samples from patients with hepatitis C virus and in the peripheral blood of individuals infected with human immunodeficiency virus, demonstrating its in vivo relevance [21]. Further mechanistic studies by the same group employed CRISPR/Cas9 to edit the promoter and second exon on a genomic level. These studies revealed that such alterations impaired the induction of *CARINH*, *IRF1*, *GBP1*, and other ISGs following IFNα stimulation. However, direct manipulation of the mature RNA form of *CARINH* through its depletion or overexpression did not affect ISG activation, highlighting that not only the RNA transcript but also the chromatin context of the *CARINH* locus may be crucial for activating IFN-responsive promoters [20].

A study examining the role of *CARINH* in virally infected macrophages provided further mechanistic insights into its function. This research discovered a *cis*-regulatory mechanism through which *CARINH* directly influences *IRF1* transcription. Initially, it was demonstrated that *CARINH* and *IRF1* are co-upregulated in the circulation of patients infected with influenza A (IAV), severe acute respiratory syndrome coronavirus 2 (SARS-CoV-2), or human metapneumovirus (hMPV). A similar co-upregulation was also observed in human macrophages exposed to IAV, the viral mimic polyinosinic–polycytidylic acid (poly[I:C]), as well as IFNβ. Functionally, the genomic deletion of *Carinh* in mice resulted in reduction of *Irf1* levels and its downstream antiviral transcriptional program, as evidenced by impaired cytokine induction and increased viral burden in mice challenged with a sublethal dose of IAV. Chromatin Isolation by RNA Purification followed by DNA sequencing (ChIRP-seq) revealed that *CARINH* binds directly to the *IRF1* locus, indicative of *cis*-regulatory function (Figure 2) [22].

Intriguingly, viral-induced expression of *CARINH* was also observed in non-model species. In goat endometrial epithelial cells, infection with the peste des petits ruminants virus (PPRV) led to a boost of ISGs through IRF3 activation, driven by enhanced *CARINH* levels [38]. The findings highlight that *CARINH* serves as a conserved antiviral regulator across different species and functions as a powerful amplifier of IFN signaling.

### 3.2. CARINH in Autoimmunity and Chronic Inflammatory Diseases

Currently, the role of *CARINH* has been implicated in several immune-driven conditions, including inflammatory bowel disease (IBD) [23,24], psoriasis [39,40], asthma, and Chronic Obstructive Pulmonary Disease (COPD) [41,42] as well as cardiometabolic disorders [43,44,45]. Additionally, there is a broader connection between *CARINH* and loci with autoimmune diseases, such as juvenile idiopathic arthritis [46] (Figure 2).

#### 3.2.1. Inflammatory Bowel Disease

Among these areas of research, studies on IBD have provided the most detailed understanding of *CARINH*’s role in disease. Analysis of RNA sequencing data from 1821 mucosal biopsies from the Mount Sinai Crohn’s and Colitis Registry identified *CARINH* as a hub gene within a module linked to inflammation and disease severity. Further functional studies revealed that *CARINH* enhances inflammatory cytokine production in LPS-stimulated primary human myeloid cells. Interestingly, these experiments showed that *IRF1* expression was minimally affected by this treatment, suggesting that *CARINH* might promote cytokine production independently of *IRF1* in this context [24]. Mechanistic studies in *Carinh^—/—^* mice revealed that although these mice appeared phenotypically normal under steady-state conditions, they experienced significantly more severe colitis compared to their wildtype counterparts when subjected to dextran sulfate sodium (DSS) or trinitrobenzene sulfonic acid (TNBS). This increased severity was characterized by greater weight loss, colon shortening, bleeding, inflammatory cell infiltration, and crypt damage, highlighting *Carinh*’s protective role against colitis. At the molecular level, RNA-sequencing analysis of bone marrow-derived macrophages (BMDMs) from *Carinh^—/—^* mice revealed marked downregulation of *Irf1* and ISGs, including *Il18bp* and *Gbp*. Combination of RNA and chromatin immunoprecipitation assays showed that *Carinh* physically interacts with the histone acetyltransferases p300/CBP and mediates the deposition of H3k27 acetylation marks at the *Irf1* locus, driving *Irf1* transcription and subsequent downstream activation of ISGs (Figure 2). These data were strengthened by the identification of eight IBD-associated single nucleotide polymorphisms (SNPs) in the 5q31.1 locus, with rs2188962 emerging as the top candidate causal SNP (C > T), increasing the carrier’s risk of IBD by approximately 7.5% [23]. Interestingly, dysregulated crosstalk between the gut immune system and microbiota is a key driver of IBD. In this context, *Carinh*, *Irf1*, *and Il18bp* are microbiota-dependent, with their expression sustained by intestinal commensals in myeloid cells and reduced in germ-free or antibiotic-treated mice. *Carinh* deficiency disrupts microbial balance, enriching pro-inflammatory taxa like *Prevotellaceae*, and its microbiota transfer confers heightened colitis susceptibility, linking depletion of *Carinh* to defective antimicrobial defense.

#### 3.2.2. Cardiometabolic Disorders

Autoimmune disorders are known to elevate the risk of cardiovascular diseases [47]; however, the shared genetic and biological mechanisms underlying these associations remain largely elusive. Over the past decades, Genome Wide Association Studies (GWASs) have identified that most risk-related SNPs are in noncoding regions of the genome [48]. This observation highlights the significant role of noncoding regulatory elements, such as promoters and enhancers, as well as transcribed noncoding RNAs, including lncRNAs, in disease susceptibility. Notably, a comprehensive integration of large-scale GWAS datasets—including over 10,000 psoriasis cases and approximately 400,000 controls—alongside UK Biobank cardiovascular cohorts, revealed significant positive genetic correlations between psoriasis and various cardiovascular conditions, such as hypertension, coronary heart disease, coronary atherosclerosis, and heart failure. Among 13 loci encompassing 653 SNPs shared between psoriasis and cardiovascular traits, the 5q31.1 locus, which includes *CARINH*, emerged as particularly significant in these associations [39].

The analysis of genotype data from over 3000 metabolic dysfunction-associated fatty liver disease (MAFLD) cases and more than 10,000 controls, sourced from the Korean Genome and Epidemiology Study, revealed strong associations with established risk loci, such as *PNPLA3*, *SAMM50*, *PARVB*, *TM6SF2*, and *GCKR*. However, by employing new methodologies to annotate and prioritize variants and genes from these association results, including Multivariate Analysis of Genomic Annotation (MAGMA) [49] and Functional Mapping and Annotation (FUMA)’s SNP2Gene and GENE2FUNC tools [50], several novel loci were identified. Among these, *CARINH* showed significant association (*p* = 4.4 × 10^−5^), with 232 variants reported [43]. This enrichment suggests that variants at the *CARINH* locus may influence MAFLD susceptibility and underscores the idea that *CARINH* acts as a genetic hub connecting immune dysregulation, metabolic stress, and chronic inflammation across tissues.

Using weighted gene co-expression network analysis (WGCNA) on publicly available datasets [51,52], ten lncRNAs, including *CARINH*, were identified that strongly correlated with Tetralogy of Fallot (TOF), a congenital heart disease characterized by a combination of ventricular septal defect, overriding aorta, pulmonary stenosis, and right ventricular hypertrophy. The data reported suggested that *CARINH* may contribute to TOF pathogenesis through disruption of splicing machinery. While mechanistically distinct from the described inflammatory circuits of *CARINH*, these findings reinforce a broader principle: lncRNAs act as central disease mediators in cardiovascular disease [45].

#### 3.2.3. CARINH Variants as Determinants of Inflammatory Disease Risk and Therapeutic Response

GWASs and subsequent meta-analyses have linked *CARINH* to a wide array of autoimmune-related diseases. In the context of psoriasis, *CARINH* has been associated with therapeutic responses. This connection was established through two independent cohorts of psoriatic patients treated with a biosimilar of Etanercept, a TNFα inhibitor, which identified 148 good responders and 29 poor responders. A meta-analysis of 3.4 million SNPs identified 7 loci with potential associations to treatment outcomes, among which rs13166823 within the *CARINH* locus exhibited one of the strongest signals. The protective minor allele was significantly more prevalent among good responders in both cohorts, suggesting that *CARINH* expression might influence clinical responses to TNFα blockade [40].

In juvenile idiopathic arthritis (JIA), genetic associations previously reported [53] were further validated by an independent replication study [46]. Using the Australian CLARITY cohort, which includes 404 JIA patients and 676 healthy controls, 19 candidate SNPs were genotyped. Subsequent regression analyses were performed, adjusting for sex and ancestry. This analysis revealed seven loci replicated with consistent direction of effect, including the 5q31.1 region encompassing IRF1 and *CARINH*. The lead SNP at this locus, rs4705862, showed a significant association with JIA (odds ratio = ~1.27, *p* = 1.1 × 10^−5^), providing independent confirmation of prior findings. This replication study further establishes robust genetic evidence that variation at the *CARINH* locus contributes to JIA risk [46], and, considering studies demonstrating its role in IFN signaling and intestinal immune homeostasis, highlights a functional lncRNA as a mechanistic link between JIA and broader autoimmune pathophysiology.

Two large GWASs aimed at uncovering genetic risk factors for asthma and COPD identified two distinct SNPs associated with *CARINH*. The SNP rs2158101 was identified as the strongest male-specific variant linked to COPD, while rs3749833 was primarily associated with susceptibility to fixed airflow obstruction. Further functional annotation suggested these SNPs may exert regulatory effects on the expression of nearby genes that play roles in immune responses and inflammation. Subgroup analyses confirmed that these associations were independent of smoking status and age at asthma diagnosis. Additionally, phenome-wide association studies demonstrated that this locus was strongly associated with traits related to eosinophil counts, atopy, and asthma, highlighting its relevance in respiratory disease mechanisms [41,42].

More broadly, the molecular mechanisms underlying the autoimmune-associated variant rs17622517, located in an intron of the *CARINH* gene, were investigated. To this end, primary human monocytes from 134 donors stimulated with LPS were studied, and they found that rs17622517 influenced the expression of *IRF1* and subsequent function. Furthermore, perturbation of the SNP locus resulted in a decrease in *IRF1*, suggesting that this region has enhancer functions [54], confirming work by others [20]. In a transcriptomic study utilizing LPS-stimulated human HMC3 microglial cells, five upregulated lncRNAs, including *CARINH*, were identified. The induction of *CARINH* was completely inhibited by JQ1, a small-molecule inhibitor that prevents epigenetic readers from binding to acetylated histones. Correlation analysis showed that *CARINH* expression was linked to 57 protein-coding genes, which included ISGs such as *BATF2*, *RSAD2*, *IFIH1*, *IFIT3*, *TLR3*, *TRANK1*, and *GBP5*. Pathway enrichment analysis revealed associations with terms related to type I IFN signaling, antiviral defense, chemokine-mediated signaling, and the negative regulation of metabolic processes. It also pointed to the activation of cell death pathways like TRAIL signaling and necroptosis. These findings suggest that *CARINH* acts as an LPS-responsive lncRNA with roles in coordinating inflammatory responses in human monocytes and microglial cells [55].

## 4. CARINH and Cancer

IFNs are crucial mediators of antitumor immunity, exerting effects directly on cancer cells and indirectly via the immune system. When IFNs bind to their receptors, they activate the Janus kinase-signal transducer and activator of transcription (JAK-STAT) pathway. This activation triggers the transcription of hundreds of ISGs that control various cellular processes, including cancer cell proliferation, apoptosis, differentiation, survival, and migration. Additionally, IFNs orchestrate immune responses by modulating the activity of nearly all immune cell subsets, making them pivotal in tumor-immune interactions [56,57]. As expected, given its role in regulating *IRF1* expression and the IFN response, *CARINH* is implicated in various cancers (Figure 3) [25,26,27,28,29].

### 4.1. Esophageal Squamous Cell Carcinoma

The first direct evidence linking *CARINH* to cancer was observed in esophageal cancer, specifically within esophageal squamous cell carcinoma (ESCC). A transcriptomic analysis, along with the integration of lncRNA expression changes in three ESCC cell lines (KYSE30, KYSE180, and KYSE45) treated with IFNβ, identified nearly 1000 lncRNAs with altered expression levels (499 being upregulated, and 495 downregulated). Notably, among the identified lncRNA, *CARINH* was most significantly induced upon IFN stimulation. Moreover, *CARINH* levels were positively correlated with *IRF1* expression in these ESCC cell lines. Expression studies in ESCC cell lines further confirmed that *CARINH* is a genuine ISG, induced by both type I (IFNβ) and type II (IFNγ) IFNs via the canonical JAK-STAT pathway. This was demonstrated by the effect of treatment with the JAK inhibitor Ruxolitinib, which inhibited *CARINH* expression in ESCC cell lines. Further support came from publicly available ENCODE Chromatin Immunoprecipitation (ChIP) sequencing data, which showed binding of STAT proteins at the *CARINH* promoter. Subsequent ChIP-qPCR validation confirmed STAT1 occupancy at the promoter region, further reinforcing the role of *CARINH* in the IFN pathway [25].

Mechanistically, *CARINH* associates with the transcriptional co-activators Interleukin Enhancer Binding Factor 3 (ILF3) and DExH-Box Helicase 9 (DHX9), acting as a nuclear scaffold to augment binding to the IRF1 promoter, driving its transcription and subsequent IFN signaling. In turn, IRF1 binds the *CARINH* promoter to boost its transcription, suggesting a positive feed-forward regulatory loop between *CARINH* and *IRF1*. Loss-of-function experiments using short hairpin RNAs (shRNAs) revealed that knocking down *CARINH* promoted colony formation of KYSE cells, reduced cisplatin-induced apoptosis in vitro, and led to increased tumor growth in xenografts of subcutaneously injected KYSE30 cells. This growth was accompanied by elevated levels of proliferation factors such as Antigen Kiel 67 (Ki67) and Proliferating Cell Nuclear Antigen (PCNA) and a reduction in apoptotic markers. Conversely, *CARINH* overexpression suppressed proliferation, impaired colony formation, and promoted apoptosis in vitro, while tumor weights and volumes of in vivo xenografts were reduced when compared to controls [25].

Clinically, *CARINH* expression was significantly reduced in ESCC tumors compared to healthy esophageal tissues, and low expression correlated with poor clinical outcomes, consistent with a tumor-suppressive function [25]. Interestingly, analyses of The Atlas of Noncoding RNAs in Cancer (TANRIC [58]) datasets revealed positive correlations between *CARINH* and *IRF1* expression across multiple cancer types, including lung, breast, and head and neck squamous cell carcinoma [25], suggesting that this regulatory axis may extend beyond ESCC.

### 4.2. Bladder Cancer

Building on the broader association with cancer, a study of bladder cancer investigated *CARINH* as part of an epithelial-to-mesenchymal transition (EMT)-related lncRNA signature linked to cancer prognosis and disease progression. By analyzing RNA-sequencing data and paired clinical information from 402 bladder cancer patients sourced from The Cancer Genome Atlas (TCGA) [59], 525 lncRNAs were found to be correlated with a curated set of 200 EMT genes [27], derived from the gene set ‘*Hallmark Epithelial-to-Mesenchymal Transition*’ in The Molecular Signatures Database [60]. Further analysis using univariate and multivariate Cox regression, alongside Gene Set Enrichment Analysis (GSEA), refined this to a prognostic signature comprising 14 lncRNAs. Interestingly, each of these lncRNA serves as an independent prognostic factor in bladder cancer patients, and *CARINH* was among the five lncRNAs associated with unfavorable outcome [27]. Of note, the classification of these lncRNAs as “EMT-related” is based on co-expression patterns with EMT hallmark genes, without supporting functional assays or further mechanistic insights. This leaves it unclear whether *CARINH* has a direct causal role in bladder cancer progression. Consequently, the clinical significance of *CARINH* in relation to altering EMT in bladder cancer remains preliminary and necessitates direct experimental investigation.

### 4.3. Acute Lymphoblastic Leukemia

A third study extended the scope of *CARINH* to hematologic malignancies, specifically acute lymphoblastic leukemia (ALL). LncRNAs are increasingly recognized as central players in hematological cancers [61,62,63], and dysregulated lncRNA expression has been reported in ALL as well [64,65,66]. Elevated expression levels of *CARINH* were found in blood samples collected from 33 patients—subdivided in three groups: Philadelphia positive ALL, Philadelphia negative ALL, and T-cell ALL—when compared to a control group consisting of 15 healthy donors [26]. Analysis of publicly available microarray data [67] from 381 bone marrow samples, collected both at diagnosis and after eight days of remission-induction therapy, identified 52 differentially expressed lncRNAs altered in ALL, including *CARINH*. *CARINH* expression levels significantly reduced after 8 days of remission-induction therapy [26]. Notably, *CARINH* was connected through an extensive competing endogenous (ce)RNA network comprising 6 lncRNA, 17 miRNAs, and nearly 200 mRNAs, evaluated using miRcode, Targetscan, miRTarBase, and Cytoscope software [68,69]. This network is associated with inflammation via TNFα, as well as hypoxia, DNA repair, apoptosis, metastasis, and proliferation [26]. Within a larger ceRNA network, *CARINH* levels were associated with, miR-31, miR-144, miR-194, miR-216a, miR-375, miR-490, and their individual mRNA targets [26].

Albeit exploratory, this study lays the groundwork for understanding the potential role of *CARINH* in ALL. However, the contribution of *CARINH* to hallmark mechanisms of leukemogenesis, such as differentiation blockade, stemness, or therapy resistance, were not directly investigated, nor were clinical correlations with patient outcomes assessed. The findings highlight *CARINH* as a therapeutic candidate of interest across ALL subgroups. The predicted ceRNA interactions also suggest that *CARINH* intersects with pathways known to be deregulated in ALL, warranting further validation in leukemic cells.

### 4.4. Breast Cancer

A cross-ancestry GWAS meta-analysis conducted with data from the Breast Cancer Association Consortium, which included women of African (9241 cases and 10,193 controls) and European ancestry (122,977 cases and 105,974 controls), pinpointed four loci linked to overall breast cancer risk. Among these, the SNP rs2522057, located at the 5q31.3 locus, was strongly associated with the expression of *IRF1*. This SNP maps approximately 15 kb downstream of *IRF1*, within an intron of *CARINH*. Functional annotation using the 25-state chromatin model indicated that rs2522057 aligns with active enhancer and promoter regions, characterized by H3K4me1 and H3K27ac histone modifications [28]. These findings suggest that rs2522057 and its associated gene *CARINH* could be key mediators of breast cancer susceptibility by influencing the expression of *IRF1* as well as other nearby genes such as *RAD50* and *SLC22A5*, with all three genes implicated in breast cancer [70,71,72,73]. The presence of rs2522057 as a risk locus in both African and European populations and its proximity to oncogenic genes bolsters its candidacy as a causal variant driving cancer progression. Further in vitro and in vivo studies are necessary to confirm this causal role and to uncover the mechanisms through which rs2522057 may regulate *IRF1*, *RAD50*, and *SCL22A5* expression via *CARINH*, ultimately affecting cancer development.

In another study, a comprehensive transcriptomic analysis was conducted to identify immune-related lncRNAs associated with breast cancer prognosis and immune infiltration. Using a publicly available microarray dataset of 327 breast cancer cases [74], researchers correlated lncRNA expression with immune-related genes and applied multivariate Cox regression to establish a six-lncRNA prognostic signature. This signature, which included *CARINH*, effectively stratified patients into high- and low-risk groups that demonstrated significantly different outcomes. The prognostic accuracy of the model was further enhanced when combined with clinical variables. Patients with higher expression levels of *CARINH* tended to fall into the low-risk group, which correlated with better survival rates and increased infiltration of CD8^+^ T-cells. This finding suggests that *CARINH* could serve as a favorable prognostic biomarker in breast cancer by modulating antitumor immune responses [29]. While the observations in these two patient studies point to a putative regulatory role for *CARINH* in breast cancer, further in vitro and in vivo functional studies are essential to validate its causal influence and to uncover the mechanisms through which *CARINH* impacts carcinogenesis.

## 5. Perspective

### 5.1. Expression of CARINH in Other Cells and Cell Types

*CARINH* exhibits broad expression across human tissues and cell types, as evidenced by publicly available data from the NIH Genotype-Tissue Expression (GTEx) Project (V8, August 2019; 17,382 RNA-sequencing samples from 948 donors spanning 54 tissues) (Figure 4a) [75]. Elevated expression is particularly evident in the lung, spleen, colon, and whole blood, consistent with the reported role of *CARINH* in antiviral responses to respiratory infections and in the pathogenesis of inflammatory bowel disease [20,21,22,23,24]. It is plausible that specific cell populations, particularly inflammatory subtypes within these organs, exhibit differential *CARINH* expression in response to environmental cues; however, tissue-wide analyses of this cellular heterogeneity and its functional implications have not yet been undertaken on this scale.

In parallel with the GTEx initiative, large-scale single-cell transcriptomic resources have been generated for the mouse, comprising nearly 100,000 cells profiled across 20 different organs and tissues. This compendium, called Tabula Muris [76], enables high-resolution, controlled comparisons of gene expression across shared cell types, such as immune populations residing in distinct anatomical compartments, thereby providing a powerful framework for dissecting tissue-specific and context-dependent transcriptional programs. Like its human counterpart, analysis of the Tabula Muris dataset demonstrates that the mouse ortholog *Carinh* is broadly expressed across diverse tissues (Figure 4b,c). In contrast to the bulk tissue profiles provided by GTEx, Tabula Muris offers single-cell resolution, readily accessible through its online platform (https://tabula-muris.sf.czbiohub.org/, accessed on 20 November 2025). This resource provides a unique opportunity to interrogate *Carinh* and other noncoding transcripts at the single-cell level across multiple organs of interest.

### 5.2. Subcellular Localization of CARINH

The subcellular localization patterns of lncRNA transcripts allow for the classification of these molecules into five distinct groups: (I) lncRNAs forming one or two prominent foci within the nucleus; (II) those with large nuclear foci accompanied by scattered single molecules throughout the nucleus; (III) mostly nuclear lncRNAs without forming any foci; (IV) lncRNAs present in both the cytoplasm and nucleus; and (V) lncRNAs primarily found in the cytoplasm. Notably, about 55% of lncRNAs are predominantly expressed in the nucleus, falling into categories I through III [77]. The subcellular localization of *CARINH* has not been extensively researched, yet functional studies have primarily indicated nuclear localization and chromatin-associated roles [22,23,25], implying that *CARINH* is likely localized in the nucleus, particularly in response to certain stimuli or in specific cell types. However, these studies do not fully rule out the presence of small cytoplasmic fractions or the possibility of cell type-specific redistribution. Additionally, in silico subcellular localization tools and databases like lncLocator [78] and RNALocate [79] suggest a more uniform distribution throughout the cell (Figure 5). Future research is expected to provide new insights into the subcellular localization of *CARINH*, which may change based on cell type, cellular activation, or cell cycle stage, reflecting the inherent dynamic nature of RNA, particularly lncRNAs.

### 5.3. RNA Modifications Driving CARINH Expression and Function

RNA modifications, such as *N*^6^-methyladenosine (m^6^A) methylation, are increasingly recognized as important regulators of RNA function. These modifications play a crucial role in modulating the accessibility of protein-binding motifs, which can influence various aspects of RNA biology. For example, m^6^A methylation can affect the export of RNA molecules to the cytoplasm and has implications for their targeting to ribosomes, thereby impacting translation efficiency. By altering these processes, m^6^A and similar modifications contribute to the dynamic regulation of gene expression and cellular function [80]. As our understanding of these modifications expands, it becomes clear that they are integral to the complex regulatory networks governing RNA behavior. To date, *CARINH* has not been described to undergo RNA methylation or to be linked to the expression of m^6^A-related genes. However, in silico prediction by sRAMP (a sequence-based RNA adenosine methylation site predictor) [81] for the three human splice variants of *CARINH* suggest distinct RNA modification profiles for each variant (Figure 6). This indicates the potential for splice variant-specific roles of *CARINH* driven by differences in RNA methylation. Future experimental studies are necessary to confirm these in silico predictions and explore the potential functional consequences.

### 5.4. Coding Potential of CARINH

Despite traditionally being considered noncoding, recent studies have identified thousands of functional micropeptides produced from small open reading frames (smORFs) hidden within lncRNA transcripts [82,83,84,85,86]. Recent high-throughput advancements for evaluating the translational capacity of these smORFs have uncovered that these micropeptides may contribute significantly to most, if not all, biological pathways [87,88].

Analysis by the NIH’s ORF finder (https://www.ncbi.nlm.nih.gov/orffinder/) of the three *CARINH* splice sequences identified over 50 different putative ORFs, producing a range of amino acid lengths. Further assessment of protein-coding potential using the Coding Potential Calculator (CPC) [89] revealed that two out of the tree splice variants have a coding potential score indicating the presence of one or more translatable ORFs (Figure 7a). Among these is a putative peptide consisting of 188 amino acids, found in variant *ENST00000337752*.6, which has the highest reliability score determined by CPC. For conceptual purposes, we analyzed this putative peptide using I-TASSER [90] to predict protein structure and function, as well as DeepLoc [91] to predict subcellular localization. These analyses suggest likely nuclear localization (Figure 7b), a peptide with seven alpha helices (Figure 7c), and a potential function in metal ion binding (Figure 7d). Though this is primarily an academic exercise, these analyses offer an in silico pipeline for exploring novel ncRNA-derived peptides. However, a more in-depth analysis and functional evaluation of other putative peptides in *CARINH* are needed to uncover any further biological insights. It is important to note that previous BLAST analysis of a 149-amino-acid peptide revealed only a single low-homology hit, and codon substitution frequency scores were negative [25]. Meanwhile, experiments using a mini-circle reporter system showed that neither the human nor the murine *CARINH* locus had any ribosome binding activity, a finding confirmed by Ribo-seq analysis [23]. These experiments provide compelling evidence confirming *CARINH* as a noncoding transcript, overriding any in silico predictions to the contrary.

### 5.5. Chromatin- and Protein-Binding Abilities of CARINH

A combination of RNA pull-down assays with mass spectrometry, DNA sequencing, and targeted immunoprecipitation has yielded detailed insights into the molecular mechanisms through which *CARINH* regulates *IRF1* expression, including evidence for a potential feed-forward loop involving both molecules [22,23,25]. Interestingly, disruption of the *CARINH* locus in HeLa cells demonstrated that transcriptional activity at this site alone may be sufficient to promote ISG expression, highlighting a potential enhancer-like role for *CARINH* [20]. On a chromatin level, *CARINH* binding is shown to associate with the DNA at approximately 232 loci, with the highest enrichment observed at the *IRF1* locus [22]. This includes its binding to *IL18BP*, a gene previously shown to be regulated by *CARINH* through IRF1 [23], hinting at the potential for direct control by *CARINH* as well. Furthermore, transcription factor motif enrichment analysis using HOMER [92] of these 232 loci revealed that *CARINH* may binds to 71 loci in conjunction with DNA-binding motifs associated with the RUNX transcription family (Figure 8a). The RUNX family, known for various functions, can alter type I IFN signaling [93,94]. This suggests that *CARINH* might mediate target gene expression by competing or cooperating with RUNX family members.

Physical interactions between *CARINH* and several transcriptional co-activators, such as ILF3, DHX9, and p300/CBP, have been identified [23,25]. However, the specific mechanisms through which these RNA–protein interactions occur on the molecular level remain unclear. One important structural feature that supports RNA–protein interactions is tandem G-rich sequences, which form G-quadruplexes. These G-quadruplexes are secondary RNA structures that can serve as docking sites for proteins [95]. Analysis of *CARINH* sequences using the Quadruplex forming G-Rich Sequences (QGRS) mapper [96], an algorithm designed to identify and provide details about the composition and distribution of potential QGRS in nucleotide sequences, revealed between 10 and 20 unique QGRS (Figure 8b), unveiling a potential molecular mechanism, as has been described for other lncRNAs [97,98]. Alternatively, CatRAPID [99] can be utilized to explore potential RNA-binding domains within *CARINH*. This method identified predicted binding domains containing general RNA recognition motifs as well as motifs for DEAD/DEAH box helicases, a family that includes DHX9, known to interact with *CARINH* [25] (Figure 8c). In turn, linking these identified motifs in *CARINH* back to proteins pointed to several members of the heterogeneous nuclear ribonucleoproteins (hnRNPs), a family of functionally diverse proteins involved in mRNA stabilization, as well as transcriptional and translational regulation [100]. Notably, hnRNPs have been reported to interact with lncRNAs that function as transcriptional mediators for immune response genes [101,102] (Figure 8d). To date, experimental evidence has identified molecular mechanisms centered on the regulatory control of *IRF1* expression by *CARINH* through association with several transcriptional co-activators [23,25]. However, an examination of *CARINH’s* chromatin-binding loci suggests the possibility of alternative mechanisms for gene regulation by *CARINH* [22]. These mechanisms are likely to be uncovered through further in silico analysis as well as functional experimentation.

### 5.6. CARINH as a Competing Endogenous RNA

The lncRNA–miRNA–mRNA axis is a critical regulatory network controlling gene expression at multiple levels. LncRNAs often sequester miRNAs, and through these interactions, lncRNAs modulate miRNA-mediated post-transcriptional repression of mRNAs, creating a dynamic interplay. This axis plays an essential role in various biological processes such as development, differentiation, immune modulation, and disease progression. Dysregulation of the lncRNA–miRNA–mRNA network has been implicated in cancers, cardiovascular diseases, and inflammatory conditions [15,17,103]. Despite its potential, understanding this axis remains challenging due to tissue- and cell-specificity and intricate interactions beyond direct RNA–RNA competition.

Studies in leukemia suggest that *CARINH* interacts with several miRNAs, including miR-31, miR-144, miR-194, miR-216a, miR-375, and miR-490, thereby influencing the translation of their downstream targets. However, when using the prediction algorithm miR-Target2 (miRDB [104]), which identifies interactions based on shared features of miRNA binding and target suppression, only a potential interaction with miR-194, a miRNA known to promote anti-tumor immunity in pancreatic cancer [105], was detected (Appendix A). This points to a possible secondary layer of lncRNA–miRNA crosstalk within the ceRNA networks investigated in ALL. Interestingly, among the miRDB-predicted miRNA interactions with *CARINH* were several miRNAs that were linked to innate immunity and cancer biology, such as miR-342-5p [106,107], miR-423-5p [108,109], miR-766-3p [110,111], and miR-4516 [112], among many others (Appendix A). These potential interactions should be interpreted with caution, as *CARINH* is predominantly localized to the nucleus and associated with nuclear functions [22,23,25]. This raises questions about its capacity to act as a ceRNA. Consequently, any proposed *CARINH*–miRNA interactions must be validated through rigorous approaches, such as luciferase reporter assays, site-directed mutagenesis, and functional perturbation studies, to definitively establish their biological relevance and impact on downstream targets [15,16].

## 6. Conclusions

In this review, we have highlighted *CARINH*, a conserved long noncoding RNA that is upregulated upon viral infection, as well as upon stimulation with IFNs [20,21,22,25,38]. *CARINH* has been shown to control the expression of its neighboring gene *IRF1* in a *cis-*regulatory manner, likely through binding of p300/CBP, driving the activation of ISGs and subsequent immune response [22,23]. Due to its effects on the IFN-driven immunity, *CARINH* has been implicated viral infections [20,21,22,38], immune and auto-immune diseases [23,24,39,40,41,42], and cardiometabolic disorders [43,44,45]. Interestingly, many SNPs have been detected at the *CARINH* locus that were associated with disease, and although direct correlation between expression of *CARINH* and these SNPS have not been established, these data underline the importance of this locus in linking genetic variation to immune regulation.

In cancer research, *CARINH* has been found to interact with transcriptional co-activators DHX9, ILF3 to regulate the expression of *IRF1*, with its loss contributing to tumor progression [23,25]. Additionally, *CARINH* is part of an extensive ceRNA network that is involved in various cancer-related pathways, such as metastasis and cell proliferation [26]. Patients’ studies further suggest that *CARINH* is included in several lncRNA signatures that may be predictive of clinical outcomes [27,28,29], highlighting its potential as a future biomarker in cancer. However, the precise link between elevated levels of *CARINH* and either poor or favorable prognosis remains unclear and is likely cancer dependent. This will be a focal point of future scientific exploration within this field.

As lncRNAs continue to be recognized as crucial orchestrators of genome architecture and gene expression regulation, the scientific community is increasingly optimistic about targeting RNA for therapeutic purposes. Previously deemed undruggable, RNA has now become the focus of a growing class of pharmacological interventions, notably including antisense oligonucleotides (ASOs), short polymers comprising 15 to 21 nucleotides [113]. To date, at least thirteen ASO-based treatments have received clinical approval for conditions like familial hypercholesterolemia and chylomicronemi [114], underscoring their promise in the development of therapies targeting lncRNAs. ncRNAs have thus become significant targets for therapeutic innovation, with substantial efforts directed at the development of oligonucleotide-based treatments. Currently, progress in programs targeting miRNAs is more advanced than those focusing on lncRNAs; however, analogous methodologies utilizing synthetically modified ASOs are being harnessed to address these differences [115,116]. GapmeRs, which are short DNA ASOs flanked by RNA mimics, and small interfering RNAs (siRNA) have been effectively employed to target specific lncRNAs such as *CARINH* [20,22,23,25]. GapmeRs, in particular, are noteworthy for their strong affinity to target RNAs, enabling them to bind efficiently before being degraded by RNAseH1 [117], thereby minimizing off-target effects [118,119]. As such, gaining a deeper understanding of the molecular mechanisms underlying *CARINH*’s functions, along with the capability to effectively target this lncRNA, will be crucial for unlocking its potential in both therapeutic and diagnostic applications.

## Figures and Tables

**Figure 1 ncrna-11-00079-f001:**
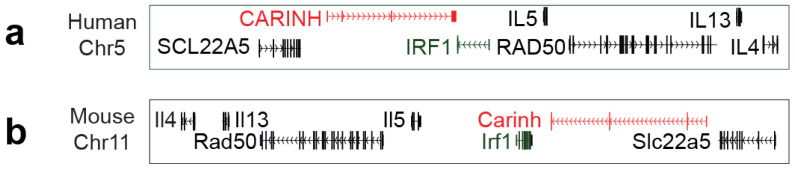
(**a**,**b**) Schematic representation of human (**a**) and mouse (**b**) IRF1/Irf1 loci (green) showing lncRNAs *CARINH* and *Carinh* (red). Arrowheads indicate direction of transcription.

**Figure 2 ncrna-11-00079-f002:**
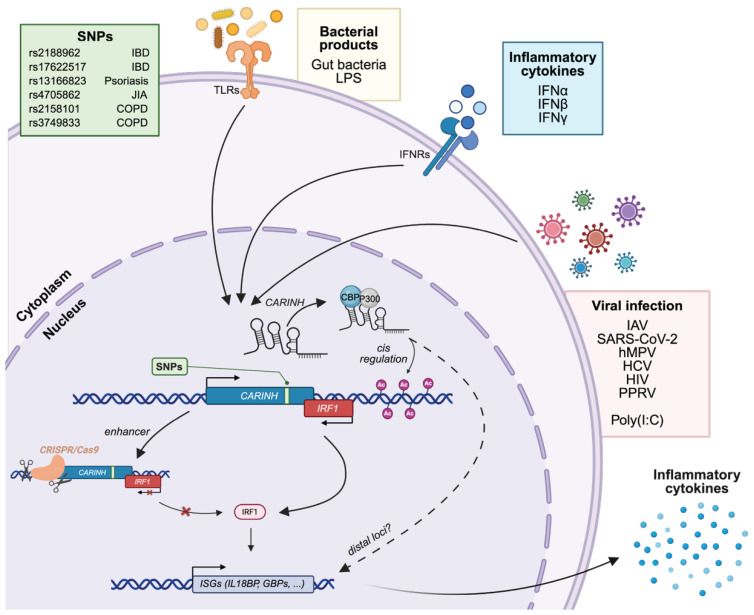
***CARINH* is a central regulator of innate and autoimmune inflammatory responses.** *CARINH* expression is triggered by diverse environmental and cellular stimuli, including inflammatory cytokines, bacterial and viral products, and viral infections. *CARINH* acts as a cis-regulatory lncRNA that promotes *IRF1* transcription through interaction with chromatin regulators such as CBP/p300. Elevated *IRF1* levels drive the expression of ISGs contributing to antiviral and inflammatory responses. Genetic variants (SNPs) within the *CARINH* locus are associated with several immune-mediated disorders, including inflammatory bowel disease (IBD), psoriasis, juvenile idiopathic arthritis (JIA), and chronic obstructive pulmonary disease (COPD), highlighting its relevance in immune homeostasis. CRISPR/Cas9-mediated disruption of the *CARINH* locus impairs *IRF1* transcription and downstream ISG induction, underscoring the enhancer-like regulatory function of *CARINH* in controlling innate immune gene expression. Solid arrows indicate established mechanisms, dashed lines indicate mechanisms that are postulated based on published data.

**Figure 3 ncrna-11-00079-f003:**
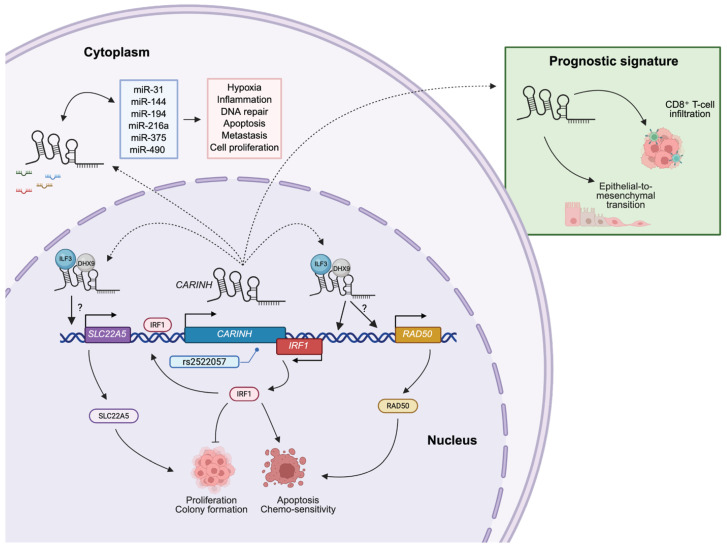
***CARINH* modulates cancer-associated pathways, thereby attenuating tumor development.** *CARINH* expression is found in a variety of cancers, such as esophageal cancer, bladder cancer, leukemia, and breast cancer. *CARINH* can exert control within its locus, such as on *IRF1*, *RAD50*, and *SLC22A5*, which in turn control key pathways in cancer growth. In the cytoplasm, *CARINH* acts as a competing endogenous RNA on a set of microRNAs associated with pathways linked to tumor cell proliferation and metastasis. *CARINH* has been shown to be part of the prognostic signature driving epithelial-to-mesenchymal transition and CD8^+^ T-cell infiltration. Solid arrows indicate established mechanisms, dashed lines indicate mechanisms that are postulated based on pulished data.

**Figure 4 ncrna-11-00079-f004:**
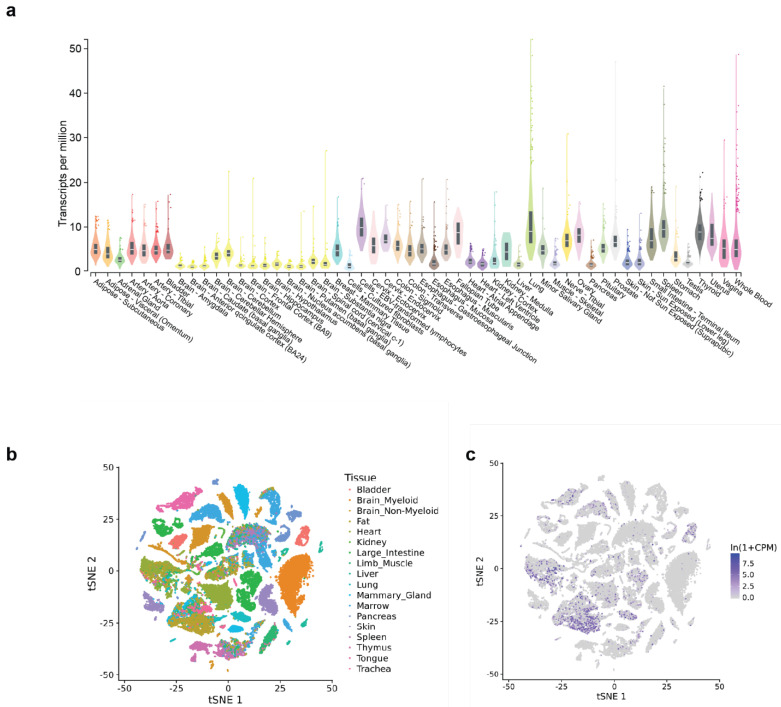
(**a**) Violin plot detailing the expression in transcripts per million of *CARINH* in 54 different human tissues. Data is provided by the National Institutes of Health Adult Genotype Tissue Expression Project. Data is derived from RNA-sequencing of 17,382 samples, 948 donors. (**b**) t-Stochastic neighbor embedding (t-SNE) visualization of all cells collected by Tabula Maris, colored by organ. (**c**) t-SNE visualization from (**b**) overlaid with dots showing level of *Carinh* expression (color scale) within each organ. Data is derived from Tabular Maris. This data includes 53,760 cells from 20 tissues from 8 mice.

**Figure 5 ncrna-11-00079-f005:**
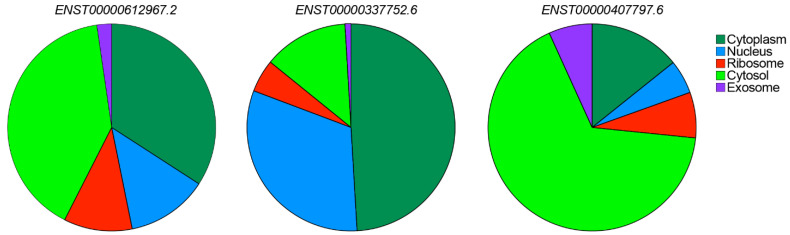
LncLocator prediction of subcellular localization of the three human variants of *CARINH*. Colors indicate different cell compartments.

**Figure 6 ncrna-11-00079-f006:**
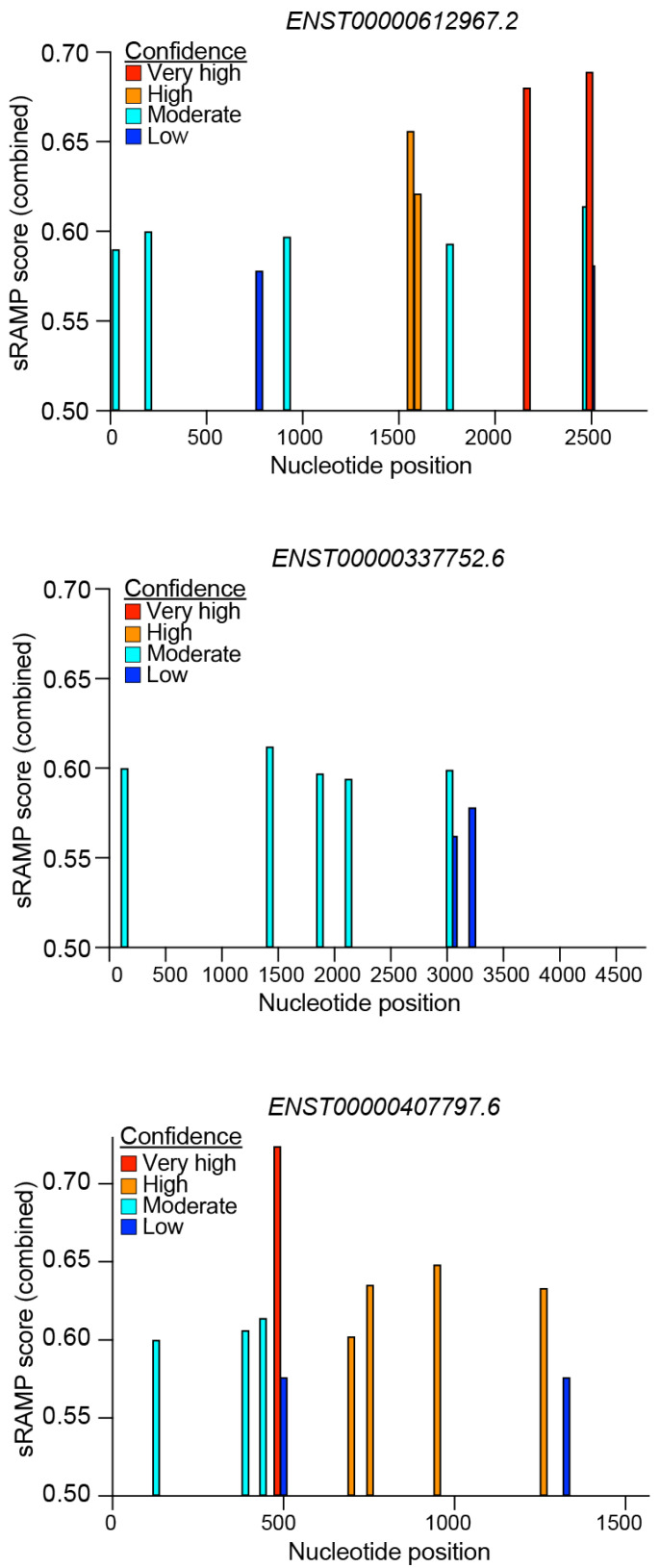
sRAMP prediction of methylation sites within the three human variants of *CARINH*. Colors indicate confidence scores.

**Figure 7 ncrna-11-00079-f007:**
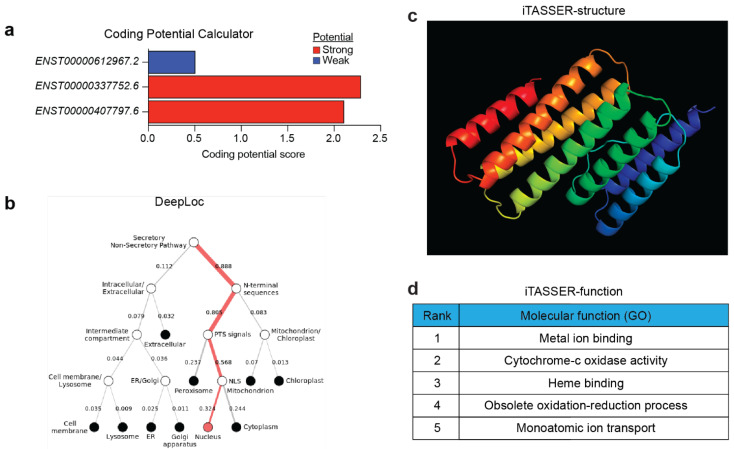
(**a**) Bar graph representing coding potential of three *CARINH* splice variants, as calculated by Coding Potential Calculator. Colors indicate strong/weak potential. (**b**) Hierarchical tree of predicted subcellular localization of a putative peptide present in *CARINH* as predicted by DeepLoc. (**c**) iTASSER predicted structure of a putative peptide present in *CARINH*. (**d**) iTASSER predicted molecular function of a putative peptide present in *CARINH*.

**Figure 8 ncrna-11-00079-f008:**
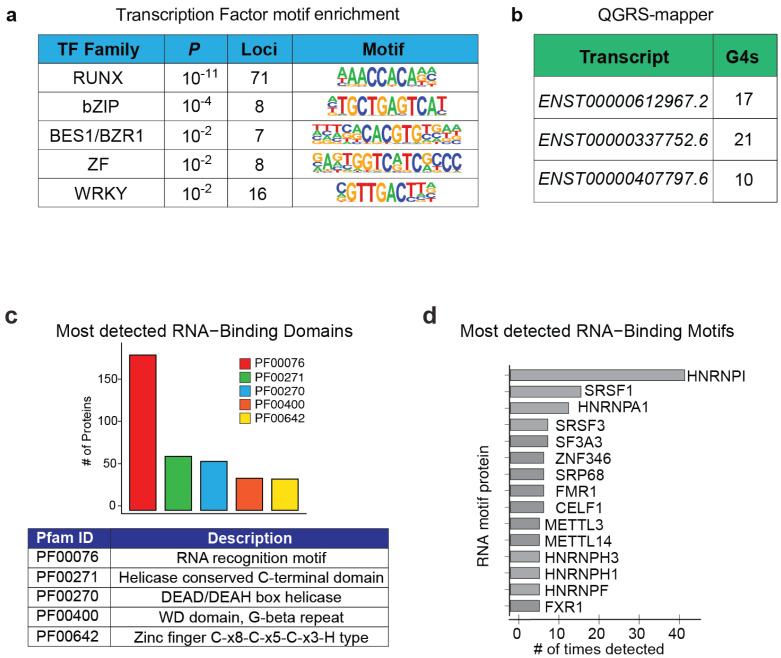
(**a**) HOMER analysis of regions bound by *CARINH*, showing transcription factor families with lowest *p*-value, number of loci, and transcription factor binding motif. *p*-values calculated using a binomial test. (**b**) Table denoting number of unique G-quadruplexes (G4s) found using QGRS mapper in each of the *CARINH* splice variants. (**c**) Identified RNA-binding domains in proteins predicted to bind *CARINH* predicted by catRAPID. (**d**) List of proteins containing most RNA-binding motifs that are predicted to bind *CARINH* detected by catRAPID.

## Data Availability

Not applicable.

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
