# Peer review of "CARINH, an Interferon-Induced LncRNA in Cancer and Inflammation"

_ncrna, 2025, doi:10.3390/ncrna11060079_

Round 1

Reviewer 1 Report

Comments and Suggestions for Authors

The manuscript represents a comprehensive and well-organized review of the long noncoding RNA CARINH and its biological relevance in immunology and oncology. The authors nicely summarize current knowledge on CARINH’s genomic context, conservation, and regulatory role in interferon-mediated innate immunity. The review integrates findings from different diseases , including viral infection, chronic inflammatory disorders, and multiple cancer types. The mechanistic description is particularly detailed, highlighting cis-regulation of IRF1, chromatin interactions with p300/CBP, and the impact of CARINH deficiency in relevant mouse models. Molecular aspects are examined in depth, including SNP-based disease associations, predicted RNA–protein and RNA–miRNA interactions, possible post-transcriptional modifications such as m6A, and subcellular localization patterns.

Overall, the manuscript provides a clear and exhaustive summary of current research about CARINH.

This review is timely and highly relevant to the readership of Noncoding RNA. The writing is clear, with well-curated literature and helpful figures. The correlation of genetic and functional evidence across inflammatory and malignant contexts clearly illustrates the growing recognition of CARINH as a regulatory hub connecting innate immunity and disease pathogenesis. The authors successfully address multiple mechanistic layers, including chromatin remodeling, interferon-stimulated gene regulation, and ceRNA roles. The inclusion of discussions on RNA modifications, coding potential, and cellular distribution provides additional depth and an advanced perspective that will be valuable to the field.

In conclusion, this manuscript represents a well-constructed review that constitutes a useful reference for researchers investigating lncRNAs in immune regulation and cancer biology. The work merits publication in its current form.

Reviewer 2 Report

Comments and Suggestions for Authors

In this review the authors elaborated the function of non-coding RNA CARINH during various cellular signaling process such as viral Infection and innate immunity and further discussed the role of CARINH in cancers including esophageal squamous cell carcinoma, acute lymphoblastic leukemia, bladder and breast cancer. Overall, the review is written well with particular focus on the subject in discussion. The review covers broad range of topics on the function of CARINH including Cardiometabolic Disorders and inflammation. However, the review needs few rearrangements of the topics to have a proper flow of reading. Moreover, additional details on CARINH would benefit the readers. Therefore, following are my recommendations.

  • A brief introduction about the general mechanism of ncRNA synthesis and its regulation with particular focus on any regulatory steps unique to CARINH would help
  • Section 5.5. on “Chromatin- and Protein-Binding Abilities of CARINH” can be appropriately discussed under section 2. CARINH: Nomenclature, Genomic Location and Synteny.
  • LncRNAs are less conserved across species owing to rapid evolution (PMID: 26773003; PMID: 36596869). However, some lncRNA families are closely conserved among species such as MALAT. The authors mentioned CARINH is conserved across species, if CARINH belongs to a family of lncRNA that are conserved among species either by function or structure, it is recommended to present a diagram demonstrating the sequence motif of CARINH at least in the IRF-1 binding motif among different mammalian species such as human, mouse, rat and monkey.
  • Also, details on different isoforms of CARINH such as CARINH-V1, CARINH-V2, and CARINH-V3 is missing in Section 2. CARINH: Nomenclature, Genomic Location and Synteny.
  • The authors listed several miRNA interaction partners of CARINH, however one the major mode of gene regulation of lncRNA is to act as “sponges" for miRNAs, preventing them from binding to their target messenger RNAs (mRNAs). Discussion on the three-way axis of gene regulation via lncRNA-miRNA-mRNA is missing in the review.
  • Moreover, the authors should include one more column listing prominent mRNA targets of miRNA in table 1.
  • Figure 3- How miR-31, miR-144 and other miRNAs depicted in the cartoon regulate hypoxia, inflammation, apoptosis and metastasis needs explanation in the text. Specifically, the mRNA targets of the listed miRNA can provide more insight to the mechanism of gene regulation by CARINH in these pathways.

Reviewer 3 Report

Comments and Suggestions for Authors

This is a timely and excellent review that summarizes current studies on lncRNA-CARINH, an interferon-induced long noncoding RNA positioned at the intersection of innate immunity, inflammation, and cancer. The paper is well organized, and the Perspective section provides valuable insights that will serve as a useful resource for the field. The work is suitable for publication after minor revisions to address the following points:1. Consider moving the full Table 1 to the Supplementary Materials and keeping a summarized version in the main text. 2. The manuscript would benefit from a dedicated Conclusion section summarizing the key functions and mechanistic insights of CARINH, as well as outlining open questions and future directions.
